

# Factors associated with quality of life among newly diagnosed acute ischemic stroke patients: a community-based case-control study

Fatima Ibrahim Abdulsalam[1], Prapatsorn Somsri[2], Piyapong Papitak[2], Kittipod Tussanabunyong[2], Wisit Chaveepojnkamjorn[3] and Nitikorn Phoosuwan[1,4]

[1] Community Health, Faculty of Public Health, Kasetsart University, Muang, Sakonnakhon, Thailand
[2] Sakon Nakhon Provincial Hospital, Sakhonnakhon, Thailand
[3] Epidemiology, Mahidol University, Bangkok, Thailand
[4] Uppsala University, Uppsala, Sweden

Corresponding author
Nitikorn Phoosuwan,
nitikorn.ph@ku.th

## ABSTRACT

**Background:** Stroke burden is expected to increase and surviving a bout of stroke may leave one with a chronic or disabling outcome decreasing significantly the quality of life of the sufferer. The study aimed to explore the association between quality of life (QoL) in stroke and non-stroke individuals in a predominantly agricultural community.

**Method:** A community-based case-control study was conducted among 154 cases aged 18 and 75 diagnosed with incident stroke. Controls were 554 non-stroke individuals ($n = 554$) recruited from a community where cases resided. Cases and controls were matched for age, gender and residential area. Using a self-reported questionnaire of the World Health Organization Quality of Life (WHOQOL-BREF), socio-demographic characteristics and lifestyle habits were measured in association with QoL. Statistical analyses included multivariable logistic regression models, adjusted odds ratio (aOR) and 95% confidence interval (CI).

**Results:** Significant predictors associated with low-medium QoL were having a larger waist circumference (aOR = 1.619, 95%CI [1.003–2.612]) and being a farmer (aOR = 2.976, 95%CI [1.143–7.750]) but having a current smoking habit and being male were at lesser odds with low-medium QoL (aOR = 0.381, 95%CI [0.191–0.757]) and (aOR = 0.564, 95%CI [0.323–0.985]) respectively. In all domains, women were at a higher risk of low-medium QoL while in physical health and environment domains, it was having a larger waist circumference. In both physical health and psychological domains, being married was protective to low-medium QoL so also being age 39 or younger and having a higher education in social relationship domain.

**Conclusion:** Waist circumference, occupation, smoking habit and gender are associated with low-medium QoL. Addressing the influence of such factors could create an additional therapeutic line in the primary prevention of stroke in at-risk populations.

# INTRODUCTION

Globally, stroke is said to be the second largest cause of global death and disability-adjusted life-years (DALYs) making it a major global health challenge (*Johnson et al., 2019*). According to a report by the 2016 Global Burden of Diseases, Injuries, and Risk Factors Study (GBD), from 1990 to 2016, the global age-standardized mortality rate and DALY rate seemed to decrease but stroke cases increased sharply all over the world causing about 5.5 million deaths and 116.4 million DALYs (*Johnson et al., 2019*). In the next few decades, the stroke burden is expected to increase. In Asia, the incidence of stroke varies between 116 and 483/100,000 per year, and stroke prevalence among the Association of Southeast Asian Nations (ASEAN) countries also varies (*Suwanwela & Poungvarin, 2016*). In Thailand, there is an estimate of about 80% ischemic stroke to 20% haemorrhagic stroke (*Suwanwela & Poungvarin, 2016*). A stroke is a neurological malfunction attributed to insufficient blood supply to the brain (*Albers, Marks & Lansberg, 2018*). Ischaemic brain injury (or ischaemic stroke) could lead to loss of neurologic function due to blockage of a cerebral vessel that imparts a specific area of the brain and if not irreversibly injured, evidence-based treatments can unfasten the obstruction restoring blood flow to the affected area of the brain (*Albers, Marks & Lansberg, 2018*; *Goyal et al., 2015*). In addition, lacunar infarcts are the ischemic stroke subtype with the best functional prognosis and the percentage of lacunar infarcts increases with age (*Arboix et al., 2010*).

Surviving a bout of a stroke may leave one with a chronic or disabling outcome (*Carod-Artal & Egido, 2009*) and traditionally, these outcomes are measured by a comprehensive approach to the construct of quality of life (QoL) which covers several functioning domains like physiological and biological factors, psychiatric symptoms and general health perception (*CDC, 2021*; *Wilson & Cleary, 1995*). By highlighting the importance of stroke management in several areas, this construct developed by the World Health Organization (WHO) QoL group tries to evaluate the complexity of individuals' physical health, social relationships, psychological state, personal beliefs, and level of independence to salient features of the environment (*Kjellström, Norrving & Shatchkute, 2007*; *WHOQoL Group, 1994, 1995*). Studies on the association of QoL and stroke risk have shown that after a stroke, the quality of life of the sufferer decreases significantly (*Chen et al., 2019*; *Lavados et al., 2021*), others show that some factors that are related to poor or low quality of life in stroke patients include older age (*Ranawaka et al., 2012*), gender (*Serda et al., 2015*), level of education (*Mahesh et al., 2018*), marital status (*Jaracz et al., 2002*), level of dependence or disability (*Ranawaka et al., 2012*; *Sturm et al., 2004*), and clinical severity of stroke (*Ramos-Lima et al., 2018*). It is also suggested that quality of life is an important predictor of conditions like haemodialysis, heart failure, diabetes and myocardial infarction although it is frequently assessed as a brain injury outcome (*Tsai et al., 2010*; *Venskutonyte et al., 2013*). However, in a case-control study, stroke predicted by psycho-physical stress could have a causal link between QoL and endovascular brain injury (*Egido et al., 2012*). In other

words, what is known to date includes the significant impact of stroke on QoL, with various factors such as older age, gender, education level, marital status, and stroke severity being associated with poor QoL outcomes; however, the influence of these factors in rural, agricultural populations is less understood. Another suggests that a strong predictor for the risk of stroke is found in people with a lower baseline QoL, so much so that in the study, when comparing participants in the highest quartile, those in the lowest baseline QoL quartile had an increased risk of stroke by 20% (*Shams et al., 2017*). This shows a bidirectional relationship between QoL and stroke, hence it is safe to say that QoL may be both a predictor and outcome of stroke. However, in this study QoL is treated as an outcome variable.

In stroke studies, QoL warrants more research attention providing significant clinical and policymaking implications for strategizing stroke risk prevention and after-care programs. Considering the huge disease burden and prevalence of stroke, there is an urgent and crucial need to identify and manage risk factors for timely prevention. Stroke-predictive factors often vary in different studies, especially in rural areas where they may be influenced by geography and ethnicity. This study aims to explore the relationship between quality of life (QoL) in individuals with and without stroke in a primarily agricultural community, using a case-control method to investigate these risk factors more thoroughly. The study is exploratory, seeking to understand the connection between low QoL in stroke and non-stroke individuals by identifying predictors of low QoL post-stroke within this community.

## MATERIALS AND METHODS

### Study design and study area

This was a case-control design study (ratio 1:4) paired for age, sex and place of residence as they might be seen as confounders. It was conducted between February to August 2022 in a northeastern agricultural region called Sakon Nakhon province in Thailand. With a population density of about 76.02 persons/km$^2$, Sakon Nakhon province has a current population of about 1.2 million (*Provincial Office, 2022*; *Thailand Interior Ministry, 2020*).

### Case recruitment

Cases were participants admitted to the stroke unit of Sakon Nakhon Provincial Hospital with a clinical diagnosis of incident stroke and aged between 18 and 75 years. According to modified Trial of Org 10172 in Acute Stroke Treatment (TOAST) criteria (*O'donnell et al., 2010*), and the International Statistical Classification of Diseases and Related Health Problems–10th revision (ICD-10 I630-639) of the Thailand R506 national disease surveillance system (*Bureau of Epidemiology, 2022*); participants were assessed for competence to answer the study questionnaire and if unable, the help of a family member (or caregiver) was sought. To ensure a low concordance between the responses from a participant and those provided by the family member (or caregiver), a pilot study was conducted making any such participant incapacity an exclusion criterion. Inclusion criteria were (1) Thai ethnicity; (2) ability to use Thai as a primary language; (3) 18 years of age and above; (4) onset of stroke before hospital admission and being first time admitted

with ICD-10 codes I630-639 for less than seven days. Exclusion criteria were: (1) refusal to participate in the study, and (2) reported having cognitive impairment, and (3) reported to have been previously diagnosed with stroke (ICD-10 codes:1630-639).

## Control recruitment

Controls were non-stroke individuals matched with cases for residential area, age, and gender. Inclusion criteria were (1) living for more than one year, and (2) not being diagnosed with ICD-10 codes I630–I639 in the last year. Control individuals were identified by the principal investigator. If a control did not respond, was ineligible due to a history of stroke or refused to participate, another matched by age, gender and living in the neighbourhood was invited as a control.

## Data collection and measurements

Quality of life is the sum of a range of objectively measurable life conditions experienced by an individual (*Felce & Perry, 1995*). To assess the QoL, a short version of the original World Health Organization Quality of Life (WHOQOL-100) developed by WHO was used (*WHO, 1998*). This is a Thai-translated version of the WHOQOL-BREF (*Mahatnirundkul, Tuntipivatanaskul & Pumpisanchai, 1998*), a self-reported questionnaire that assesses QoL in four different domains. Having a total of 26 items (24 items for physical (seven), psychological (six), social (three) and environmental (eight) domains), one item is for general QoL and the last for health-related QoL. Participants were asked to rate their HRQOL in the past 2 weeks and the item scores were rated on a 5-point scale ranging from 1 = "low, negative perception" to 5 = "high, positive perception". Due to a difference in the number of items for each domain, domain scores were calculated by computing the mean score of items within each domain and then multiplying by 4 to commensurate with the scores used in the WHOQOL-100. With a Cronbach's α coefficient of 0.84, the Thai WHOQOL-BREF version used in the present study expressed a good internal consistency. According to Thai WHOQOL-BREF, a total raw score of 26–60 indicates low QoL, 61–95 indicates medium QoL and 96–130 indicates high QoL (see Additional File 1) (*Mahatnirundkul, Tuntipivatanaskul & Pumpisanchai, 1998*).

Socio-demographic and clinical characteristics including age, sex, marital status, education level, main occupation, place of residence and medical history were obtained by trained research assistants. By standard protocols and techniques, anthropometric measures including weight, height, waist and hip circumference of participants were also recorded. Body mass index (BMI) was computed as weight (kg) divided by height squared (m$^2$). Our study categorized adults as underweight (BMI < 18.5 kg/m$^2$), normal weight (18.5 ≤ BMI < 24 kg/m$^2$), overweight (24 ≤ BMI < 30 kg/m$^2$), and obese (BMI ≥ 30 kg/m$^2$) (*WHO, 2011*). Adult waist circumference was categorized according to WHO-recommended population-specific values (*i.e.*, <80 cm for women and <90 cm for men of Asian origins) (*WHO, 2011*). Smoking status was categorized into non-smoker (never smoked or have had <100 cigarettes in their lifetime), quit smoker (not smoking at the time of the survey but have a history of smoking at least 100 cigarettes in their lifetime) and current smoker (smoke daily during the time of the survey)

(*Schoenborn & Adams, 2010*). Alcohol consumption was also categorized into none (never had alcohol in their lifetime), quit alcohol (not taking alcohol at the time of the study) and currently take alcohol (at the time of the survey) (*Platz et al., 1999*).

Using the same standardised format in cases and control, data were collected using a questionnaire and face-to-face interviews. For cases, interviews were conducted in the hospital during the first week of hospital admission following the stroke incident.

### Sample size estimates and Statistical analysis

The sample size was calculated for a confidence level α error of 5%, a power of 80%, an estimated proportion of exposure of controls of 32.4% (*Gravel & Béland, 2005*), an odds ratio (OR) of 1.8 and a case-control ratio of 1:4 resulting in a total number of 150 cases and 600 controls. In the end, we had a total response rate of 94.4% (*i.e.*, 708 participants).

Using the IBM SPSS Statistics for Windows Version 22.0 (SPSS ver. 22.0; Armonk, NY, USA) computer program for all statistical analyses, descriptive statistics were presented as frequency and mean ± SD. The outcome variable was dichotomized into 1 = case, and 0 = control. Univariable logistic regression was used to determine the associations between the predictor and outcome variable and significant variables were further chosen for multivariable logistic regression analysis to control for possible confounding effects and to assess the separate effects of each independent variable on the outcome variable. The backward stepwise LR (likelihood ratio) method was used for entering variables and the goodness of fit of the model was checked by the Hosmer-Lemeshow test. The strength of the statistical association was measured by adjusted odds ratios (aOR) and 95% confidence intervals (CI).

### Ethical consideration

The study was approved by the ethics committees of Sakon Nakhon Hospital (COA026/2564) and Kasetsart University (Kucsc.HE-64-008), following the Declaration of Helsinki principles. Participants received oral and written information and provided written consent, with the assurance of voluntary participation and the right to withdraw at any time. Authors had no involvement in participants' care. Data was coded for anonymity, and participants' privacy and confidentiality were strictly maintained.

## RESULTS

In a total of 708 participants, there were 154 (21.8%) stroke cases and 554 (78.2%) controls. Men were 254 (35.9%) while women were 454 (64.1%), the mean age of all participants was 59.24 (±9.52) years and BMI was 24.37 (±4.24) kg/m$^2$ with a significant difference between cases and control ($p$ = 0.002, 95%CI [−7.498 and −1.748]). The majority were Buddhist (94.9%), married (74.6%), had a primary school level of education or less (75.1%) and farmers by occupation (71.8%). Most had no smoking or alcohol consumption habits (79.4% and 63.7%) respectively or a family history of paralysis or cardiovascular disease (92.8% and 96.3%). See Table 1.

The predictors associated with low-medium QOL levels in both cases and controls include having a waist circumference above the WHO cut-off (aOR = 1.619, 95%CI

**Table 1 Socio-demographic characteristics of study participants stratified by cases and control.**

| Characteristics | | All (n = 708) | Cases (n = 154) | Controls (n = 554) | t | $\chi^2$ | p-value | 95% CI | |
|---|---|---|---|---|---|---|---|---|---|
| | | | | | | | | Lower | Upper |
| Age (years (Mean ± SD)) Min = 21, Max = 75 | | 59.24 ± 9.52 | 58.54 ± 11.06 | 59.44 ± 9.05 | −1.033 | | 0.302 | −2.599 | 0.807 |
| BMI (kg/m$^2$ (Mean ± SD)) Min = 14.04, Max = 46.67 | | 24.37 ± 4.24 | 23.68 ± 4.46 | 24.57 ± 4.16 | −2.310 | | 0.002 | −7.498 | −1.748 |
| Waist circumference (according to WHO standard (n (%))) | | | | | | | | | |
| Female | <80 cm | 104 (22.9) | 25 (24) | 79 (76) | | 9.802 | 0.002 | | |
| | ≥80 cm | 350 (77.1) | 41 (11.7) | 309 (88.3) | | | | | |
| | Mean (SD) | 88.04 (11.10), Min = 60, Max = 119 | 84.09 (11.37), Min = 68, Max = 114 | 88.71 (10.92), Min = 60, Max = 119 | | | | | |
| Male | <90 cm | 164 (64.6) | 57 (34.8) | 107 (65.2) | | 0.002 | 0.960 | | |
| | ≥90 cm | 90 (35.4) | 31 (34.4) | 59 (65.6) | | | | | |
| | Mean (SD) | 85.44 (9.82), Min = 60, Max = 120 | 84.43 (8.75), Min = 65, Max = 110 | 85.97 (10.33), Min = 60, Max = 120 | | | | | |
| Sex (n (%)) | | | | | | | | | |
| Male | | 254 (35.9) | 88 (57.1) | 166 (30) | | 38.694 | <0.001 | | |
| Female | | 454 (64.1) | 66 (42.9) | 388 (70) | | | | | |
| Religion (n (%)) | | | | | | | | | |
| Buddhism | | 672 (94.9) | 149 (96.8) | 523 (94.4) | | 1.378 | 0.241 | | |
| Christainity | | 36 (5.1) | 5 (3.2) | 31 (5.6) | | | | | |
| Marital status (n (%)) | | | | | | | | | |
| Not Married | | 180 (25.4) | 31 (20.1) | 149 (26.9) | | 2.909 | 0.088 | | |
| Married | | 528 (74.6) | 123 (79.9) | 405 (73.1) | | | | | |
| Education level (n (%)) | | | | | | | | | |
| Primary school or less | | 532 (75.1) | 112 (72.7) | 420 (75.8) | | 9.284 | 0.010 | | |
| Secondary school | | 137 (19.4) | 26 (16.9) | 111 (20) | | | | | |
| Higher education (diploma or degree) | | 39 (5.5) | 16 (10.4) | 23 (4.2) | | | | | |
| Occupation (n (%)) | | | | | | | | | |
| Farmers | | 508 (71.8) | 76 (49.4) | 432 (78) | | 54.153 | <0.001 | | |
| Self employed | | 104 (14.7) | 34 (22.1) | 70 (12.6) | | | | | |
| Civil servant | | 26 (3.7) | 13 (8.4) | 13 (2.3) | | | | | |
| Others (unemployed, housewives, VHVs) | | 70 (9.9) | 31 (20.1) | 39 (7) | | | | | |
| Smoking habit (n (%)) | | | | | | | | | |
| Never smoked | | 562 (79.4) | 93 (60.4) | 469 (84.7) | | 47.687 | <0.001 | | |
| Quit smoking | | 65 (9.2) | 22 (14.3) | 43 (7.8) | | | | | |
| Current smoker | | 81 (11.4) | 39 (25.3) | 42 (7.6) | | | | | |
| Alcohol consumption (n (%)) | | | | | | | | | |
| Never had alcohol | | 451 (63.7) | 76 (49.4) | 375 (67.7) | | 17.787 | <0.001 | | |
| Quit alcohol | | 87 (12.3) | 28 (18.2) | 59 (10.6) | | | | | |
| Currently take alcohol | | 170 (24) | 50 (32.5) | 120 (21.7) | | | | | |

| Characteristics | All (n = 708) | Cases (n = 154) | Controls (n = 554) | t | $\chi^2$ | p-value | 95% CI | |
|---|---|---|---|---|---|---|---|---|
| | | | | | | | Lower | Upper |
| Family history of paralysis (n (%)) | | | | | | | | |
| No | 657 (92.8) | 143 (92.9) | 514 (92.8) | | 0.001 | 0.974 | | |
| Yes | 51 (7.2) | 11 (7.1) | 40 (7.2) | | | | | |
| Family history of cardiovascular disease (n (%)) | | | | | | | | |
| No | 682 (96.3) | 149 (96.8) | 533 (96.2) | | 0.101 | 0.751 | | |
| Yes | 26 (3.7) | 5 (3.2) | 21 (3.8) | | | | | |

Note:
   BMI, body mass index; CI, confidence interval; SD, standard deviation; VHV; village health volunteers; WHO, world health organization.

**Table 2 Univariable and multivariable analyses of risk factors associated with low-medium QoL in stroke participants.**

| Category | Sub-category | All n (%) | Cases n (%) | Control n (%) | Univariable cOR (95%CI) | Multivariable aOR (95%CI) |
|---|---|---|---|---|---|---|
| Age (years) | 70 or more | 106 (19.7) | 25 (23.6) | 81 (76.4) | 1 | – |
| | 60–69 | 171 (31.8) | 43 (25.1) | 128 (74.9) | 0.919 [0.52–1.62] | – |
| | 50–59 | 178 (33.1) | 39 (21.9) | 139 (78.1) | 1.100 [0.62–1.95] | – |
| | 40–49 | 70 (13) | 19 (27.1) | 51 (72.9) | 0.828 [0.42–1.66] | – |
| | 39 or less | 12 (2.2) | 5 (41.7) | 7 (58.3) | 0.432 [0.13–1.48] | – |
| Mean (SD) = 59.54 (9.66), Min = 26, Max = 75 | | | | | | |
| BMI (kg/m$^2$) | Normal weight | 214 (39.9) | 57 (26.6) | 157 (73.4) | 1 | – |
| | Obese | 54 (10.1) | 9 (16.7) | 45 (83.3) | 1.815 [0.83–3.95] | – |
| | Overweight | 230 (42.8) | 50 (21.7) | 180 (78.3) | 1.307 [0.85–2.02] | – |
| | Underweight | 39 (7.3) | 15 (38.5) | 24 (61.5) | 0.581 [0.29–1.19] | – |
| Mean (SD) = 24.49 (4.35), Min = 14.04, Max = 46.67 | | | | | | |
| Waist circumference (cm) | Standard | 205 (38.2) | 70 (34.1) | 135 (65.9) | 1 | – |
| | Greater than standard | 332 (61.8) | 61 (18.4) | 271 (81.6) | 2.304 [1.54–3.44]* | 1.619 [1.01–2.61]* |
| Gender | Female | 343 (63.9) | 59 (17.2) | 284 (82.8) | 1 | – |
| | Male | 194 (36.1) | 72 (37.1) | 122 (62.9) | 0.352 [0.24–0.53]* | 0.564 [0.32–0.99]* |
| Marital status | Not married | 134 (25) | 27 (20.1) | 107 (79.9) | 1 | – |
| | Married | 403 (75) | 104 (25.8) | 299 (74.2) | 0.725 [0.45–1.17] | – |
| Education level | Primary school or less | 413 (76.9) | 102 (24.7) | 311 (75.3) | 1 | – |
| | Secondary school | 96 (17.9) | 17 (17.7) | 79 (82.3) | 1.063 [0.58–1.93] | – |
| | Higher education (diploma or degree) | 28 (5.2) | 12 (42.9) | 16 (57.1) | 0.405 [0.15–1.11] | – |
| Occupation | Civil servant | 21 (3.9) | 11 (52.4) | 10 (47.6) | 1 | – |
| | Self employed | 68 (12.7) | 24 (35.3) | 44 (64.7) | 2.017 [0.745–5.43] | 1.214 [0.42–3.54] |
| | Farmers | 386 (71.9) | 67 (17.4) | 319 (82.6) | 5.237 [2.14–12.83]* | 2.976 [1.14–7.75]* |
| | Others (unemployed, VHVs, housewives) | 62 (11.5) | 29 (46.8) | 33 (53.2) | 1.252 [0.47–3.37] | 0.497 [0.17–1.48] |
| Smoking habit | Never smoked | 428 (79.7) | 80 (18.7) | 348 (81.3) | 1 | – |
| | Quit smoking | 48 (8.9) | 19 (39.6) | 29 (60.4) | 0.351 [0.19–0.66]* | 0.573 [0.26–1.29] |
| | Current smoker | 61 (11.4) | 32 (52.5) | 29 (47.5) | 0.208 [0.12–0.36]* | 0.381 [0.19–0.76]* |

(Continued)

| Table 2 (continued) | | | | | | |
|---|---|---|---|---|---|---|
| Category | Sub-category | All n (%) | Cases n (%) | Control n (%) | Univariable cOR (95%CI) | Multivariable aOR (95%CI) |
| Alcohol consumption | Never had alcohol | 356 (66.3) | 69 (19.4) | 287 (80.6) | 1 | – |
| | Quit alcohol | 62 (11.5) | 23 (37.1) | 39 (62.9) | 0.408 [0.23–0.73]* | 0.800 [0.38–1.66] |
| | Currently take alcohol | 119 (22.2) | 39 (32.8) | 80 (67.2) | 0.493 [0.31–0.79]* | 0.776 [0.44–1.35] |

Notes:
* Statistically significant at 0.05 level.
cOR, crude Odds Ratio from univariable analysis; aOR, adjusted Odds Ratio from multivariable analysis; BMI, Body Mass Index; CI, confidence interval; QoL; quality of life; SD, Standard Deviation; VHVs, village health volunteers.

[1.003–2.612]), being male (aOR = 0.564, 95%CI [0.323–0.985]), being a farmer (aOR = 2.976, 95%CI [1.143–7.750]), and currently having a smoking habit (aOR = 0.381, 95%CI [0.191–0.757]). See Table 2.

For the physical health domain, having a waist circumference higher than the WHO cut-off is a predictor of low-medium QoL (aOR 1.697, 95%CI [1.04–2.76]) while protective factors include being male (aOR 0.406, 95%CI [0.25–0.66]) and married (aOR 0.391, 95% CI [0.21–0.72]). The protective factors for psychological domain were also being male (aOR 0.354, 95%CI [0.19–0.65]) and married (aOR 0.347, 95%CI [0.15–0.77]). Similarly, for the social relationships domain, being 39 years or less (aOR 0.273, 95%CI [0.08–0.98]), male (aOR 0.350, 95%CI [0.21–0.57]) and having higher education (aOR 0.237, 95%CI [0.06–0.91]) were protective of low-medium QoL. The predictive factor for the environment domain was having a waist circumference higher than the WHO cut-off (aOR 2.097, 95%CI [1.13–3.88]) while being a male (aOR 0.373, 95%CI [0.20–0.69]) was protective of low-medium QoL. See Table 3.

## DISCUSSION

Our study suggested that in all domains, males were at lesser odds of having low-moderate QoL and having a bigger waist circumference than the WHO set criteria values in both physical health and environment domains was a risk factor for low-medium QoL. Also, being married was protective to having a low-medium QoL in both physical health and psychological domains, likewise, being 39 years old or less and having a higher education in the social domain. As for the overall QoL, significant predictors associated with low-medium QoL were having a bigger waist circumference and being a farmer, but having a current smoking habit and being male were protective of having low-medium QoL. In Thailand, there is an ongoing challenge for healthcare professionals to improve or maintain an optimal quality of life for stroke patients as it has been reported that recurrent stroke is a unique feature more common in the Southeast Asia (SEA) region compared to other countries (Khan et al., 2011).

According to our results, males were at lesser odds of having low-medium QoL. Several factors do influence QoL after stroke but reports show that women have it worse for up to a year after a stroke occurrence (Bushnell et al., 2014; Singhpoo et al., 2012). This is observed more in the mental and physical domains (Gray et al., 2007), and it could be likely due to

**Table 3 Analyses of factors associated with low-medium QoL for each domain in stroke participants.**

| Category | Sub-category | Physical health | | Psychological | | Social relationships | | Environment | |
|---|---|---|---|---|---|---|---|---|---|
| | | cOR (95% CI) | aOR (95% CI) | cOR (95% CI) | aOR (95% CI) | cOR (95% CI) | aOR (95% CI) | cOR (95% CI) | aOR (95% CI) |
| Age (years) | 70 or more | Reference | | | | | | | |
| | 60–69 | 0.80 [0.40–1.61] | – | 0.960 [0.38–2.46] | – | 1.066 [0.54–2.11] | 1.339 [0.65–2.75] | 1.067 [0.43–2.67] | 1.280 [0.49–3.33] |
| | 50–59 | 0.784 [0.40–1.55] | – | 0.656 [0.27–1.58] | – | 1.047 [0.54–2.02] | 1.203 [0.60–2.41] | 0.669 [0.29–1.55] | 0.614 [0.26–1.48] |
| | 40–49 | 1.104 [0.44–2.78] | – | 0.933 [0.29–2.99] | – | 0.758 [0.35–1.64] | 0.838 [0.38–1.87] | 0.716 [0.25–2.07] | 0.853 [0.28–2.58] |
| | 39 or less | 0.395 [0.10–1.54] | – | 0.167 [0.04–0.68] | – | 0.235 [0.07–0.79]* | 0.205 [0.06–0.73]* | 0.273 [0.08–0.98]* | 0.288 [0.07–1.12] |
| Gender | Female | Reference | | | | | | | |
| | Male | 0.371 [0.24–0.59]* | 0.406 [0.25–0.66]* | 0.30 [0.17–0.54]* | 0.354 [0.19–0.65]* | 0.314 [0.20–0.50]* | 0.350 [0.21–0.57]* | 0.327 [0.19–0.59]* | 0.373 [0.20–0.69]* |
| BMI (kg/m$^2$) | Normal weight | Reference | | | | | | | |
| | Obese | 0.742 [0.34–1.60] | – | 0.957 [0.35–2.65] | – | 0.706 [0.32–1.54] | – | 1.052 [0.36–3.09] | – |
| | Overweight | 0.761 [0.47–1.24] | – | 0.736 [0.39–1.40] | – | 0.782 [0.48–1.27] | – | 0.818 [0.44–1.51] | – |
| | Underweight | 1.740 [0.58–5.26] | – | 0.663 [0.22–2.05] | – | 1.449 [0.52–4.01] | – | 0.861 [0.29–2.57] | – |
| Waist circumference (cm) | Standard | Reference | | | | | | | |
| | Greater than standard | 1.975 [1.25–3.11]* | 1.697 [1.04–2.76]* | 2.332 [1.30–4.17]* | 1.810 [0.97–3.37] | 2.103 [1.34–3.30]* | 1.619 [0.99–2.66] | 2.690 [1.52–4.76]* | 2.097 [1.13–3.88]* |
| Marital status | Not Married | Reference | | | | | | | |
| | Married | 0.442 [0.24–0.80]* | 0.391 [0.21–0.72]* | 0.338 [0.15–0.75]* | 0.347 [0.15–0.77]* | 0.677 [0.40–1.14] | – | 0.511 [0.26–1.02] | – |
| Education level | Primary school or less | Reference | | | | | | | |
| | Secondary school | 0.552 [0.30–1.03] | – | 1.043 [0.45–2.41] | – | 0.628 [0.32–1.23] | 0.581 [0.29–1.18] | 0.522 [0.24–1.14] | – |
| | Higher education[a] | 0.658 [0.17–2.54] | – | 0.608 [0.15–2.44] | – | 0.266 [0.08–0.94]* | 0.237 [0.06–0.91]* | 0.598 [0.11–3.17] | – |
| Occupation | Civil servant | Reference | | | | | | | |
| | Self employed | 0.278 [0.03–2.33] | – | 0.287 [0.03–2.52] | – | 1.282 [0.30–5.45] | – | 1.057 [0.19–5.90] | – |
| | Farmers | 0.280 [0.04–2.18] | – | 0.331 [0.04–2.62] | – | 1.000 [0.27–3.73] | – | 0.713 [0.15–3.37] | – |
| | Others[b] | 0.256 [0.03–2.17] | – | 0.208 [0.02–1.82] | – | 0.975 [0.23–4.13] | – | 0.675 [0.12–3.74] | – |
| Smoking habit | Never smoked | Reference | | | | | | | |
| | Quit smoking | 1.190 [0.55–2.57] | – | 0.487 [0.21–1.16] | – | 0.801 [0.37–1.72] | – | 0.789 [0.30–2.10] | – |
| | Current smoker | 2.330 [0.96–5.64] | – | 2.065 [0.60–7.150] | – | 1.527 [0.66–3.56] | – | 2.505 [0.73–8.57] | – |
| Alcohol consumption | Never had alcohol | Reference | | | | | | | |
| | Quit alcohol | 1.007 [0.49–2.06] | – | 1.423 [0.52–3.91] | – | 1.120 [0.52–2.44] | – | 1.345 [0.44–4.11] | – |
| | Currently take alcohol | 0.611 [0.36–1.03] | – | 0.573 [0.29–1.15] | – | 0.619 [0.37–1.05] | – | 0.542 [0.29–1.03] | – |

**Notes:**
* Statistically significant at 0.05 level.
[a] Denotes having a diploma or degree certificate.
[b] Denotes unemployed, village health volunteers and homemakers.
cOR, crude Odds Ratio from univariable analysis; aOR, adjusted Odds Ratio from multivariable analysis; BMI, Body Mass Index; CI, confidence interval; QoL, quality of life.
difficulty in mobility as women may have greater muscle function limitations affecting physical recovery (*Paolucci et al., 2006*). Also, depression in correlation with QoL was reported to be more common in women (*Taylor-Piliae, Hepworth & Coull, 2013*) perhaps because they may have higher recovery expectations or worse coping or adaptation approaches. Another reason could be that in our study, the mean age of female stroke participants was higher (60.38 ± 10.21) than male stroke participants (57.76 ± 11.53) and QoL is affected by age increase. Howbeit in some countries, the life satisfaction of stroke survivors was found to be higher among women (*Baumann et al., 2012*). Large waist circumference due to a central fat distribution indicates an excess burden of ill health (*Lean, Han & Seidell, 1998*). In our study, those with larger waist circumferences were twice at higher odds of having low-medium QoL in the physical health and environmental domains compared to those below WHO cut-offs. Among the anthropometric measures analysed in a recent study, only waist measurements showed a correlation with QoL, particularly in the physical domain (*Tozetto et al., 2021*). Reduction in physical functioning directly affects health perception and performing rigorous activities and studies have observed that carrying body weight around the midsection poses a risk for non-communicable diseases (NCDs) compared to other parts of the body (*Lean, Han & Seidell, 1998*). As waist circumference reflects central and total adiposity (*Lean, Han & Morrison, 1995*), a possible reason why BMI is not always a precise health indicator in terms of body fat is, that it doesn't differentiate between lean mass and fat mass unable to tell where most body weight is located. Also observing Tables 2 and 3 of our study, there was no relationship between BMI and QoL.

In both physical health and psychological domains, being married was a protective factor against having a low-medium QoL. Higher QoL scores were obtained by those who are married compared to those unmarried, perhaps because they are more likely to be supported by their spouses and family members consequently improving stroke disability recovery. This is similar to some studies (*Han et al., 2014*; *Jaracz et al., 2002*) but in contrast to others (*Shahbazi & Ali, 2018*) as well. In the social relationship domain of QoL, being 39 years old or less was protective against having low-medium QoL (Table 3) which was in accordance with a recent study conducted in Poland (*Tvaronavičienė et al., 2021*), the study suggested that for youths, a factor that is paramount for a positive perception of QoL is the social relationship domain, particularly family relationships. For them, this indicator characterizes a person's QoL denoting the social orientation of modern youths. This same perception could also apply to youths in Thailand. In addition, those who have a higher education were at lesser odds of obtaining low-medium QoL scores and this is comprehendible as participants' level of education may affect treatment compliance and management guidelines and a better understanding of the stroke recovery process. Besides, higher education may earn higher income and better occupation.

As for the overall QoL results of our study, having a larger waist circumference and being a farmer were the observed risk factors. Compared to civil servants who are monthly salary earners, farmers were about three times as likely to have a low-medium quality of life. This was recently observed where the quality of life of farmers in northern Thailand is deteriorating (*Phetphum et al., 2022*). Crop farming is vulnerable to natural disasters, pests

and diseases, and dwindling price of commodities, to mention but a few; these can cause exposure risks to stress and reports have shown that farmers are at health risk of cancer, depression or suicide due to debt, poverty or family stress (*Lemarchand et al., 2016*; *Szortyka et al., 2021*). More than 70% of our participants are farmers and ~77% have primary school education or none (Table 2), perhaps this is so because in such a predominantly farming community, education fees cannot be afforded and primary and secondary school children had to drop out to help at the farm. Farming households in northern Thailand live in impoverished situations (*Sricharoen, 2019*) bringing about poor life satisfaction.

Placed as one of the top ten risks leading to disease burden by WHO, the addictive behaviour of tobacco consumption is an established risk factor for NCDs like respiratory problems, lung disease, cancer, coronary heart disease and stroke (*WHO, 2002*), particularly in disadvantaged and socially marginalized populations (*WHO, 2014*). However, our study suggested that a current smoker was at lesser odds of low-medium QoL. In SEA, some reports have shown significant relationships between smoking and health-related QoL (*Chen et al., 2015*; *Kristina et al., 2015*) with a few showing little or no association (*Funahashi et al., 2011*; *Matsushita & Matsushima, 2004*) but similar to our study, another has shown that current smokers had higher QoL (*Sagtani, Thapa & Sagtani, 2019*). Young adults have expressed social benefits of smoking which include relieving stress, relaxation, enjoyment and comfort around friends (*Aryal & Bhatta, 2015*). For such addictive behaviours, perhaps participants in this study were 'light smokers' and estimating the level of QoL differs from biological factors relating to disease and mortality rates (*i.e.*, effects of such habits on the QoL model are strictly different from a medical model). Although the results in this study suggests that light smoking could have a positive effect on QoL when compared to non-smokers, often, when positive effects of smoking are found it is due to confounding (*e.g.*, age confound due to smokers being younger). However, it does not change the fact that tobacco use at any dose, including second hand smoking, is a major risk factor for a recurrent stroke and therefore, should not be recommended. Hence, further study is needed to understand better the effect of addictive behaviours on self-perceived QoL.

## STRENGTHS AND LIMITATIONS

From our study, risk factors contributing to low-medium QoL were having a larger waist circumference and being a farmer, while current smokers were at lesser odds. This enlightens the context of QoL and ischaemic stroke participants in the northeastern Thailand region. We employed a case-control design with a 1:4 matching ratio for residential areas, gender, and age to investigate risk factors for QoL, adjust for major confounders, and enhance the test's power. Data collection during the COVID-19 pandemic also provided an opportunity to study QoL during a crisis period. However, there are several limitations. Self-reported responses on social behaviours may be prejudiced, also the non-probability sampling limits external validity. The reliance on hospital records for case identification may have excluded stroke cases managed outside the hospital setting, potentially leading to selection bias. Finally, the study did not account

for the severity of stroke, which could significantly impact the quality of life and the observed associations. This study focused on the QoL, and incorporating clinical data and research on the correlation between stroke and QoL could enhance the value of research findings and implications, such as in prediction and prognosis (*Arboix et al., 2009*).

The study presented several future directions and implications for policy, practice, and lifestyle changes. Targeted interventions are necessary to address specific risk factors such as central obesity and occupation. Reducing waist circumference through lifestyle modifications like improved diet and increased physical activity could potentially enhance the QoL among stroke survivors and at-risk populations. Given that being a farmer was associated with lower QoL, there is a need for policies and programs that provide additional support to farmers, including stress management, access to healthcare, and financial assistance to mitigate the impacts of occupational stressors and economic instability. The study also found that women are at a higher risk of low-medium QoL post-stroke, suggesting the need for gender-specific healthcare strategies that address the unique challenges faced by female stroke survivors, including mental health support and rehabilitation services tailored to improve physical recovery and overall QoL.

The bidirectional relationship between stroke and QoL underscores the importance of psychosocial interventions. Programs aimed at improving social support, marital relationships, and mental health could be beneficial in both stroke prevention and post-stroke care. Policymakers should consider integrating the study's findings into public health strategies, promoting healthy lifestyles to prevent central obesity, and providing education and resources to improve the overall QoL of stroke survivors, especially in rural and agricultural communities. The study's limitations suggest the need for further research to explore the causal relationships between the identified factors and QoL, as well as to investigate additional variables that may influence outcomes. Longitudinal studies and larger, more diverse samples could provide deeper insights into the dynamics of QoL post-stroke. For individuals, adopting healthier lifestyle practices, seeking regular medical check-ups, and utilizing available social and healthcare resources could contribute to improved QoL and stroke prevention.

## CONCLUSIONS

This study explored the factors associated with the QoL among individuals with and without stroke in a predominantly agricultural community in northeastern Thailand. Significant predictors of low-medium QoL included larger waist circumference and occupation as a farmer while being male and a current smoker were associated with higher QoL. In all domains, women are at a higher risk of low-medium QoL. In the physical health and environment domains, a larger waist circumference was a significant risk factor, whereas being married, younger (39 years or less) and having a higher education were protective factors in physical health, psychological, and social relationship domains.

The findings highlight the importance of addressing specific socio-demographic and lifestyle factors in stroke prevention and management strategies. Interventions aimed at reducing central obesity and providing support for farmers could potentially improve the QoL among stroke survivors and at-risk populations in rural communities. Additionally,

the study underscores the need for targeted psychosocial interventions to enhance QoL post-stroke, particularly for women who are at higher risk of lower QoL outcomes. By identifying and addressing these factors, healthcare professionals can better support stroke patients and contribute to more effective primary prevention strategies in rural, agricultural regions.

## ACKNOWLEDGEMENTS

The authors thank all respondents who participated in this study, and Uppsala University Sweden.

### Funding
The research was funded by Kasetsart University Research and Development Institute (KURDI) and Faculty of Public Health Kasetsart University, Thailand. The funders had no role in study design, data collection and analysis, decision to publish, or preparation of the manuscript.

### Grant Disclosures
The following grant information was disclosed by the authors:
Kasetsart University Research and Development Institute (KURDI).
Faculty of Public Health Kasetsart University, Thailand.

### Competing Interests
The authors declare that they have no competing interests.

### Author Contributions

- Fatima Ibrahim Abdulsalam analyzed the data, prepared figures and/or tables, authored or reviewed drafts of the article, checked the article for spelling and grammatical errors, and approved the final draft.
- Prapatsorn Somsri conceived and designed the experiments, performed the experiments, authored or reviewed drafts of the article, and approved the final draft.
- Piyapong Papitak conceived and designed the experiments, performed the experiments, authored or reviewed drafts of the article, and approved the final draft.
- Kittipod Tussanabunyong conceived and designed the experiments, performed the experiments, authored or reviewed drafts of the article, and approved the final draft.
- Wisit Chaveepojnkamjorn conceived and designed the experiments, performed the experiments, analyzed the data, prepared figures and/or tables, and approved the final draft.
- Nitikorn Phoosuwan conceived and designed the experiments, performed the experiments, analyzed the data, prepared figures and/or tables, authored or reviewed drafts of the article, and approved the final draft.

# PeerJ

## Human Ethics

The following information was supplied relating to ethical approvals (*i.e.*, approving body and any reference numbers):

The study was approved by the Sakon Nakhon Hospital ethics committee and ethical approval for the study protocol was obtained from the Ethics Committee of Kasetsart University (Kucsc.HE-64-008) and of Sakon Nakhon Provincial Hospital (COA026/2564).

## Data Availability

The data is available in the Supplemental File.

## Supplemental Information

Supplemental information for this article can be found online at http://dx.doi.org/10.7717/peerj.18266#supplemental-information.

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
