# Peer review of "Factors associated with quality of life among newly diagnosed acute ischemic stroke patients: a community-based case-control study"

_PeerJ, doi:10.7717/peerj.18266_

## Round 0.1 · original submission · Major Revisions

Thank you for this submission. Please respond to every reviewer suggestion and include the changes you have made in the manuscript. I note that the 5th column in Table 1 has an erroneous header -- please fix this, as it should be degrees of freedom. In addition, because you did a matched case-control study ("Controls were non-stroke individuals matched with cases for residential area, age, and gender"), you need to use conditional logistic regression to analyze your matched data, so your study will require re-analysis. Also, if you did separate analyses by gender/sex, please state this.

·

Basic reporting

The authors present the results of a community-based case-control study aimed to explore the association between quality of life (QoL) in stroke and non-stroke individuals in a predominantly agricultural community (Sakon Nakhon province, in Thailand). A total of 154 cases diagnosed with incident stroke and 554 non-stroke individuals (controls) were recruited. The authors found that waist circumference, occupation, smoking habit and gender are factors associated with low-medium QoL. The study is potentially interesting, but for a better presentation of the data the following issues need to be clarified:

Experimental design

1. The authors should mention in the Introduction that lacunar infarcts are the ischemic stroke subtype with the best functional prognosis (see data and comment on the study published in Neuroepidemiology 2010;35:231-6) Did the authors take this into account in their study protocol?

2. It would be interesting to know the frequency of the different stroke subtypes (lacunar, atherothrombotic, cardioembolic, infarcts of unusual etiology, infarcts of unknown etiology, intracerebral hemorrhage) in the study sample

3. It would be interesting if the authors included in the text some of the limitations of this study.

4. It would be interesting for the authors to discuss, as a new line of research, the correlation between the cerebral topography of the stroke and the QoL. A clinical report showed that cerebral infarcts in the territory of the anterior cerebral artery have a better prognosis than infarcts in the territory of the middle cerebral artery (see data on the study published in BMC Neurol. 2009 Jul 9; 9:30. doi: 10.1186/1471-2377-9-30). Did the authors consider this in their study?

5. A brief concluding comment on other possible lines of future research on the presented topic would be appreciated

6. Please, check references: #Chen Q et al. (line 392); #Khan M et al (line 419)

Validity of the findings

No comment

Additional comments

No comment

·

Basic reporting

The ""Materials & Methods"" section, though comprehensive, is overly lengthy and could benefit from condensation for improved readability.
The detailed definition of stroke included in the ""Case Recruitment"" section is redundant and should be moved to the Introduction. Additionally, the current citation style should be reviewed to ensure it aligns with the journal's guidelines; if APA style is not required, the Vancouver style is recommended for its brevity and clarity. The ""Ethical Considerations"" section can be significantly shortened by retaining only the essential information, such as ethical approval from Sakon Nakhon Hospital and Kasetsart University, adherence to ethical principles, and the consent process, while omitting the more detailed descriptions.

Experimental design

clear

Validity of the findings

--

Additional comments

none

Reviewer 3 ·

Basic reporting

I also think the background and aim here does not match the results presented. What is/are your actual aim/s? Are you looking at predictors? What relationship are you specifically looking at? what story are you trying to tell?

The need to assess this in rural areas can be elaborated to justify conducting the study itself. Why is it important that we assess this in this cohort? What's known and what's not known till date?

lines 70-72. I am not aware of QoL measures that collect physiological and biological factors.

Lines 129- were they first ever stroke? or did you include those with recurrent events too?

Discussion:
The paragraphs had multiple things/topics in the one making it quite difficult to read.
what are future directions or implications does your study present?? What does this mean for policy and practice change? or even lifestyle changes?

Experimental design

what did these interviews include? what information was collected? Were there additional surveys in the questionnaire pack?

did you include any clinically relevant variables even if they may not be statistically significant in the models?

Validity of the findings

Study aim in abstract does reflect the results presented in the abstract. specify that you are looking at predictors of qol post-stroke in this community. The conclusion does not match to the aim presented in the abstract.

Line 139: How does this make them controls? What % of them had cerebral events in the past? Should you be including those with no history of stroke at all for a fair comparison?

were they first ever stroke? or did you include those with recurrent events too?

Additional comments

what are the several limitations of the study?

I do think this is an important paper with aspects of it being clear to read. I hope the above comments are found useful and would help shape the paper for publication.

---

## Round 0.2 · accepted · Accept

Thank you for addressing the reviewers' comments.

·

Basic reporting

I thank the authors for addressing the issues in my initial review. I am satisfied with the additional sentences added to the manuscript and have no additional suggestions to make

Experimental design

Rigorous investigation performed.

Validity of the findings

Conclusions are well stated.

Additional comments

No additional comments.